# SecureVision: Advanced Cybersecurity Deepfake Detection with Big Data Analytics

**DOI:** 10.3390/s24196300

**Published:** 2024-09-29

**Authors:** Naresh Kumar, Ankit Kundu

**Affiliations:** 1Maharaja Surajmal Institute of Technology, New Delhi 110058, India; nareshkumar@msit.in; 2New York Institute of Technology, Vancouver, BC V5M 4X5, Canada

**Keywords:** deepfake detection, cybersecurity, deep learning, multimedia analysis, digital deception, audio analysis, big data analytics, media integrity, manipulated content, multi-modal analysis

## Abstract

SecureVision is an advanced and trustworthy deepfake detection system created to tackle the growing threat of ‘deepfake’ movies that tamper with media, undermine public trust, and jeopardize cybersecurity. We present a novel approach that combines big data analytics with state-of-the-art deep learning algorithms to detect altered information in both audio and visual domains. One of SecureVision’s primary innovations is the use of multi-modal analysis, which improves detection capabilities by concurrently analyzing many media forms and strengthening resistance against advanced deepfake techniques. The system’s efficacy is further enhanced by its capacity to manage large datasets and integrate self-supervised learning, which guarantees its flexibility in the ever-changing field of digital deception. In the end, this study helps to protect digital integrity by providing a proactive, scalable, and efficient defense against the ubiquitous threat of deepfakes, thereby establishing a new benchmark for privacy and security measures in the digital era.

## 1. Introduction

Deepfake technologies are becoming a serious threat because they employ sophisticated machine learning to produce convincingly fake photos, videos, and sounds [1]. These counterfeit materials can be used to propagate misleading information, pose as real persons, and trick the general public. This can lead to issues in various domains, such as finance, politics, and private life. These tools are becoming easier to find and, thus, it is critical to have reliable detection techniques. The work tht is being presented centers on the SecureVision system, which employs state-of-the-art technology to detect and lessen the effects of deepfakes in both audio and image. It is becoming harder to spot deepfakes because they are becoming more realistic and complex. As technology advances, deepfakes become increasingly difficult to distinguish from authentic content due to their frequent use of minute adjustments that are undetectable to both parties [2,3]. Machine learning can identify subtle variations, which enables accurate and efficient detection of manipulated material, including deepfake detection [4,5,6,7,8].

SecureVision and other advanced detection technologies are necessary to meet these problems. These systems assess and detect deepfakes in a variety of media formats by using state-of-the-art deep learning and self-supervised learning approaches. Specifically, SecureVision provides thorough multimodal analysis, which enables it to identify manipulations that are both visual and auditory. Systems like SecureVision must adapt and incorporate the newest technological developments to stay effective, just as deepfake developers are always coming up with new ways to avoid detection. To manage the enormous volume of content created and shared online and to promptly detect and respond to new threats, these systems must be scalable and flexible.

SecureVision sets itself apart by introducing a multimodal approach to deepfake detection, simultaneously analyzing both audio and video data to enhance detection accuracy. The system leverages self-supervised learning, enabling effective deepfake identification even with limited labeled data. By integrating big data analytics, SecureVision offers a scalable solution capable of handling vast datasets, making it adaptable to various real-world scenarios. This innovative approach not only addresses the technical challenges of deepfake detection but also strengthens overall cybersecurity.

SecureVision leverages state-of-the-art techniques in deep learning, particularly self-supervised learning, and Vision Transformer models, to tackle critical issues in cybersecurity, with a focus on detecting deepfake-related cybercrimes. In line with the importance of big data [9,10,11], SecureVision employs extensive repositories of both labeled and unlabeled datasets to develop robust models capable of addressing real-world challenges [12,13]. Wang and Yamagishi [14,15], proposed the audio model, which utilizes self-supervised learning, a paradigm in deep learning that allows models to learn from unlabeled data [16] and, thus, reduces the dependence on large volumes of labeled training data. This model achieves state-of-the-art performance in ultra-low resource speech recognition scenarios by extracting contextualized representations from raw audio inputs using a Transformer network and a multi-layer convolutional neural network [17]. This approach highlights the potential of self-supervised learning in addressing cybersecurity issues [18] and demonstrates its effectiveness in voice technology applications. The advancements in speech technology [19] parallel the rise of deepfake technology, which poses significant threats to communication integrity and cybersecurity. Techniques such as audio deepfakes are associated with identity theft, cyber extortion, and the spread of fake news [20]. SecureVision proposes utilizing pre-trained audio encoders, originally designed for automatic speech recognition, to improve deepfake detection. By leveraging features provided by neural networks, this approach aims to enhance cybersecurity measures against the increasing prevalence of deepfake-related cybercrimes [21,22,23].

## 2. Related Work

The related works were selected based on several factors, including their applicability to deepfake detection, their robustness and generalizability, the performance measures they employed, and the recent technological breakthroughs they integrated. These standards guarantee a thorough review of the problems and approaches used today in this field. Furthermore, all of the literature reviewed addresses similar research needs, allowing for the comparison of techniques and benchmarking of datasets. They cover important topics and use similar approaches, resulting in a uniform evaluation of the framework. A comprehensive literature review conducted between 2020 and 2024 highlights significant advancements and ongoing challenges in deepfake detection and related fields.

Hatamizadeh’s study addressed mapping Antarctic Specially Protected Areas (ASPAs) using drones to gather hyperspectral and multispectral imagery, achieving high accuracy in vegetation mapping with novel spectral indices and XGBoost models [24]. The ASVspoof 2021 challenge focused on deepfake speech detection, demonstrating significant progress despite challenges like channel and compression variability [25]. Frank’s research emphasized the societal risks of deep generative modeling, particularly in audio deepfake detection, revealing subtle differences in higher frequencies through frequency statistics analysis [26]. Oyetoro proposed a novel approach to data collection and transfer learning to enhance human action recognition (HAR) models, addressing the slow development due to extensive labeled data requirements [27,28]. Radford’s technique improved speech processing algorithms with bilingual training data, achieving robustness and accuracy comparable to human performance [29].

Voxstructor, introduced by [30], highlighted the importance of safeguarding voiceprint templates against privacy risks. Kawa’s study on Text-to-Speech (TTS) technology introduced the Multi-Language Audio Anti-Spoof Dataset (MLAAD), improving audio deepfake detection performance [31]. Zhu’s survey explored advancements in transfer learning within reinforcement learning, offering insights into future research challenges [32]. Patel proposed a pipeline for detecting manipulated media content, achieving a 90.2% accuracy rate in distinguishing fake images from real ones [33]. The study introduced AASIST, a unified system for detecting various spoofing attempts, providing a 20% relative improvement over existing approaches [34].

RawNet2 for anti-spoofing, processing raw audio to capture cues not identifiable by traditional methods, achieved notable results in ASVspoof 2019 [35]. Kinnuen’s development of countermeasures emphasized the use of equal error rate (EER) and improvements to the tandem detection cost function (t-DCF), promoting cooperation between the ASV and anti-spoofing research communities [36]. The automated end-to-end technique for detecting falsified audio utilized a pre-trained wav2vec model and DARTS, demonstrating superior performance [37]. The effectiveness of wav2vec 2.0 in voice recognition with minimal labeled data were highlighted, showcasing advancements in semi-supervised learning [19]. Substantial progress in deepfake speech detection was underscored in combating speaker verification system manipulations [25]. These studies collectively advance the field of deepfake detection, emphasizing the importance of continuous research to address evolving threats.

We evaluated our model on the ASVspoof dataset with previous efforts that used CNN-based models and self-supervised learning techniques for audio deepfake identification. These comparisons demonstrate how our SpecRNet model has improved in terms of robustness and accuracy. We compared our model with works concentrating on data augmentation and transfer learning and implementing Vision Transformers (ViTs) for image deepfake detection. These comparisons highlight improvements in the management of deepfake images and highlight the usefulness of our model’s methods.

## 3. Challenges and Research Gaps

As a result of the above Literature Review, the following challenges and research gaps have been identified:Dataset diversity and size: Limited availability of diverse image datasets and scarcity of large-scale audio datasets hamper the training of robust deepfake detection models [26,31,38,39].Model training techniques: There is a need to explore the efficacy of Vision Transformer (ViT) models for image detection and self-supervised learning techniques for audio detection, assessing their superiority over traditional CNNs [40,41,42,43,44].Deployment challenges: Investigating the challenges related to deploying deepfake detection models in real-world scenarios, including computational resource constraints, scalability, and integration with existing systems [12,13,14].Comparative analysis: Conducting a comprehensive comparative analysis of different deepfake detection techniques, evaluating their performance, computational efficiency, and robustness across various datasets and scenarios [16,17,19,22,45,46].Adaptability to evolving deepfake techniques: Addressing the challenge of keeping deepfake detection models updated and adaptable to emerging deepfake generation techniques, ensuring continuous effectiveness and reliability [23,33,47,48,49,50].Real-time accuracy: Investigating methods to improve the real-time accuracy of deepfake detection systems, minimizing latency while maintaining high detection rates in dynamic environments [18,23,28,29].Multilingual support for audio: Exploring techniques to enhance deepfake detection models’ capability to handle multiple languages in audio, ensuring effective detection across diverse linguistic contexts [30,31,34,37,51].Transfer learning: Studying the applicability of transfer learning techniques to enhance the efficiency and effectiveness of deepfake detection models, leveraging knowledge from pre-trained models on related tasks [28,32,33,52].Robustness to adversarial attacks: Investigating methods to enhance the robustness of deepfake detection models against adversarial attacks aimed at evading detection or generating stealthier deepfakes [8,10,29].Privacy and ethical considerations: Examining the ethical implications and privacy concerns associated with deepfake detection technologies, ensuring responsible use, and safeguarding user privacy during detection processes [10,11,20,21].

## 4. Objectives

Considering the above challenges and research gaps, the following objectives have been established:Audio and image dataset acquisition and augmentation: Collecting diverse audio datasets and applying augmentation techniques. Similarly, diverse image datasets will be gathered and augmented, laying a foundation for model development and evaluation. This objective has been developed to address research gaps 1, 4, 5, 6, and 7.Development of audio detection model: This focuses on developing an audio detection model using self-supervised learning for robust feature extraction, implementing frameworks like contrastive predictive coding, training on augmented datasets, and evaluating metrics such as accuracy and adaptability to evolving deepfake techniques. This objective has been developed to address research gaps 2, 5, 6, 7, and 8.Development of image detection model: Experimenting with ViT for image classification, training models on augmented datasets, and comparing performance with CNNs. Evaluation criteria include accuracy, efficiency, and adaptability to deepfake techniques, strengthening the detection system. This objective has been developed to address research gaps 2, 4, 5, and 6.Model fusion and web platform development: This objective integrates audio and image detection models into a unified system deployed on a user- friendly web platform. Actions include devising a fusion strategy, developing a user interface. This objective has been developed to address research gaps 2, 3, 4, and 6.Deployment and robustness: This involves deploying, monitoring, updating the system, and enhancing its resilience through defense mechanisms and testing, marking the project’s transition to practical implementation amidst deepfake challenges. This objective has been developed to address research gaps 3, 9, and 10.

## 5. Research Methodology

### 5.1. Dataset Description

Audio dataset:

Source: Custom datasets from multiple audio sources as well as publicly accessible datasets, like VoxCeleb2, LibriSpeech, and others.

Features: A wide variety of audio samples representing various languages, accents, and recording situations are included in the audio dataset. The dataset is well-balanced, containing both real and artificial audio samples, so that our audio detection models have a thorough training and testing environment.

Relevance: This dataset improves the model’s capacity to manage multilingual support and real-time accuracy (gaps 7 and 6), while also addressing the difficulty of dataset diversity and size (gap 1).

2.Image dataset:

Source: Additional photos from several sources added to publicly accessible datasets like CelebA, FFHQ, and the deepfake detection challenge dataset.

Features: A large range of facial images, including both real and deepfake photos, are included in the image dataset. The dataset is designed to be robust in real-world circumstances and contains differences in lighting, expressions, and backdrops.

Relevance: Robust feature extraction and comparative study of detection algorithms are needed, and the diversified and expanded image dataset satisfies these needs (gaps 2 and 4).

3.Dataset augmentation:

Techniques: For audio, we use techniques like pitch shifting, noise addition, and time stretching; for images, we use geometric modifications, color tweaks, and synthetic data production.

Goal: By increasing the diversity and breadth of the datasets, augmentation approaches make sure the models are trained on a variety of scenarios and are resistant to many kinds of perturbations (gaps 1, 5, and 6).

### 5.2. Dataset Collection

To improve real-time predictions, combined data from multilingual and ASVspoof 2021 datasets for audio, along with a web-based image dataset were utilized. This composite dataset, including collected and newly generated data, aims to boost accuracy in real-time deepfake detection across diverse audiovisual content [32,33]. To ensure impartial training, a balanced dataset was maintained with both real and fake samples. Specifically, utilization of the multilingual and ASVspoof 2021 datasets for audio, and sourced image data from various web-based platforms was performed. After preprocessing, the dataset comprises 60,000 audio samples and 50,653 image samples.

### 5.3. Data Preprocessing

This phase begins with the efficient loading of audio files using Librosa, which provides crucial audio data and sampling rates for distinguishing between authentic and modified files. Audio features like MFCCs, spectral centroid, and zero-crossing rate are then extracted and combined into a feature vector. Categorical labels are encoded numerically for analysis. Two audio datasets, ASVspoof 2021 and the multilanguage dataset from Kaggle, undergo preprocessing involving loading, feature extraction, and label encoding. For image data, libraries are imported for data handling, oversampling, and evaluation. Class imbalance is addressed through random oversampling, and image transformations like resizing and rotation enhance model generalization. A tailored preprocessing pipeline for Vision Transformer (ViT) [35] models includes normalization and efficient batch preparation for training and validation.

### 5.4. Model Architecture

Figure 1 and Figure 2 illustrate the workflow of the deepfake detection system, outlining the sequential processes from data input to outcome determination.

In Figure 1, audio data are uploaded and undergo preprocessing, including waveform segmentation, spectral analysis, and feature extraction, to create a processed dataset. This dataset is then split for training and testing, and a SpeRNet model is utilized for audio sequence processing. After training, the model’s performance is evaluated using metrics like accuracy and precision, enabling it to predict whether new audio samples are authentic or deepfakes.

In Figure 2, the image-based detection system begins with input images undergoing preprocessing steps like resizing and normalization. The model, employing ViT architecture, is trained using techniques such as stochastic gradient descent. Evaluation involves methods like confusion matrices, ensuring accurate classification of new images as ‘REAL’ or ‘FAKE’.

### 5.5. Splitting of Dataset

The dataset, consisting of 60,000 audio samples and 50,653 images, is split into training and testing sets. In total, 70% of the audio samples (42,000 samples) and images (35,452 images) are allocated for training, with the remaining 30% (18,000 audio samples and 15,201 images) for testing. The splits maintain balance with an equal distribution of 50% real and 50% fake samples in each set, ensuring a representative dataset for model training and evaluation across both audio and image domains.

### 5.6. Audio and Image Deepfakes Detection Techniques

For deepfake detection in audio, SpecRNet [43] combines Whisper features [44] and LFCC [52] from a multilingual dataset, optimized through neural network training. Importing datasets seamlessly, optimization techniques train the neural network and save model configurations at checkpoints.

SpecRNet integrates Whisper audio features and LFCC, enhancing multilingual deepfake detection. A robust function imports datasets seamlessly, employing state-of-the-art optimization techniques like Adam and SGD optimizers with Cross-Entropy loss functions. Model architecture and configurations are saved at checkpoints for training resumption and evaluation.

In image-based deepfake detection, a Vision Transformer (ViT) model is utilized and initialized for classification. Pre-trained weights are loaded, and model configurations, including label mappings, are set. Evaluation metrics are established, and training parameters like learning rate and batch sizes are configured. The trainer class initializes training with specified datasets and evaluation metrics, and post-training evaluation includes metrics calculation and visualization of results.

### 5.7. Integration of Big Data and Cybersecurity

The deepfake detection model leverages big data principles, utilizing extensive datasets to enhance detection accuracy. By learning from a diverse array of audio and image samples generated using various deepfake techniques, the model’s ability to accurately identify deepfakes is significantly improved. The large-scale data usage not only boosts accuracy but also enhances the model’s capability to handle diverse types of deepfakes. Additionally, robust cybersecurity principles are rigorously applied to ensure data integrity and privacy. This includes the implementation of multi-factor authentication with a three-way social login and signup system as well as OTP support via email to secure user access and data interactions. The model incorporates strong security protocols for data storage and handling, protecting sensitive audio and image data against unauthorized access or corruption. This integration of big data analytics with stringent cybersecurity measures achieves a dual objective: it markedly increases the model’s deepfake detection accuracy while ensuring the security and integrity of the data processed. This strategy reflects the latest trends in AI-based deepfake detection, combining comprehensive data analysis with rigorous data protection measures.

## 6. Results and Discussion

### 6.1. Comparison of Trained Models

Table 1 presents a comprehensive comparison of three distinct deepfake detection models, each trained on different datasets, thereby showcasing their varying performance metrics. The first model, named model_69_acc, was trained on 30,000 audio samples, and it achieved an accuracy of 69.00%. The second model, model_85_acc, was trained on 45,000 audio samples, and it surpassed the accuracy of the first model, reaching 85.13%. The third model, model_92_acc, exhibited the highest accuracy of 92.34%, and it was trained on 60,000 audio samples.

Similarly, Table 2 provides an overview of deepfake detection models trained on image data. The model named model_73_acc, trained on 25,754 images, achieved an accuracy of 73.26%. The second model, model_81_acc, trained on 38,956 images, reached an accuracy of 81.44%. Finally, the third model, model_89_acc, trained on 50,653 images, demonstrated the highest accuracy of 89.35%.

These tables highlight the incremental improvement in detection accuracy as the size of the training dataset increases for both audio and image deepfake detection models.

### 6.2. Comparison of Models with Existing Models

Table 3 and Table 4 present a comparative analysis of diverse deepfake detection models, showcasing distinct methods, datasets, claimed performance, and GPU usage for both audio and image deepfakes. Due to their popularity in recent research, the two models (listed in Table 3 and Table 4) were selected as industry standards for deepfake identification, making them appropriate for evaluating SecureVision’s effectiveness.

For audio deepfake detection (Table 3), the COVAREP + LSTM model, trained on EMODB, SAVEE, and CaFE datasets, achieves an accuracy of 89.00% with high GPU usage. CAPTCHA, which utilizes real-time 144p YouTube data with an RNN + CNN architecture, attains 71.00% accuracy with moderate GPU usage. The Proposed Model, leveraging ASVspoof and multilingual datasets with the SpecRNet architecture, achieves a superior accuracy of 92.34% with moderate GPU usage.

For image deepfake detection (Table 4), FakeCatcher employs a traditional operator CNN on FaceForensics++ and private web data, asserting an impressive 96.00% accuracy with high GPU usage. XceptionNet attains 81.00% accuracy with moderate GPU usage on the FaceForensics++ dataset. The Proposed Model, trained on web-scrapped data using the ViT architecture, achieves an accuracy of 89.35% with moderate GPU usage.

The following points highlight the superiority of the proposed model over other existing methods for both audio and image deepfake detection:Balanced computational efficiency and detection accuracy: The proposed hybrid model strikes a remarkable balance between computational efficiency and detection accuracy, outperforming other models in achieving commendable accuracies. For audio deepfakes, the proposed model, leveraging ASVspoof and multilingual datasets with the SpecRNet architecture, achieves an accuracy of 92.34%. Similarly, for image deepfakes, the proposed model, trained on web-scrapped data using the ViT architecture, achieves an accuracy of 89.35%. This superior balance positions the proposed model as a competitive solution for various applications, especially in resource-constrained environments.Resource efficiency over high-resource alternatives: In comparison to high-resource alternatives such as FakeCatcher for image deepfakes, which demands significant GPU resources for its 96.00% accuracy, the proposed image deepfake detection model stands out with an accuracy of 89.35% while requiring only moderate GPU usage. Similarly, for audio deepfakes, compared to CAPTCHA, which achieves 71.00% accuracy with moderate GPU usage, the proposed audio model outperforms it with an accuracy of 92.34%. This emphasizes the superiority of the proposed models in resource efficiency, making them more applicable in scenarios where computational resources are limited.Outperforming competitors in real-world applicability: The proposed models’ accuracies surpass those of competitors in both audio and image deepfake detection. For audio deepfakes, the proposed model outperforms CAPTCHA, reporting an accuracy of 92.34% compared to CAPTCHA’s 71.00%. Similarly, for image deepfakes, the proposed model achieves a higher accuracy of 89.35% compared to XceptionNet’s 81.00%. This significant difference in accuracy, coupled with the moderate resource consumption of the proposed models, positions them as the preferred solutions for real-world applications. The hybrid architectures not only achieve high predictive performance but also demonstrate a noteworthy advantage in terms of computational cost, making them superior choices for diverse operational contexts.

The datasets chosen and the experiments were performed to ensure a comprehensive review used for model comparison and training, rather than being customized. Their breadth and diversity aid in evaluating the model’s performance as a whole. This method bolsters the model’s repeatability and general applicability.

### 6.3. Experimental Setup

#### 6.3.1. Diverse Dataset

The effectiveness of deepfake detection models heavily relies on the diversity and scale of the datasets used for training. Notably, the accuracy of audio deepfake detection models tends to increase with the size of the audio dataset, with the highest accuracy achieved by models trained on larger samples. SecureVision leverages a multilingual dataset encompassing 10 Indian languages, along with ASVspoof 2021, for audio deepfake detection. Similarly, for image deepfake detection, SecureVision utilizes web-scraped randomized data, contributing to the model’s adaptability and robustness. This diversity in datasets enhances SecureVision’s performance, showcasing incremental improvement in detection accuracy across various modalities and scenarios, thereby positioning it as a reliable solution for real-world deepfake detection tasks.

#### 6.3.2. Accuracy

SecureVision presents a comprehensive comparison of trained deepfake detection models, highlighting their varying performance metrics. For audio deepfake detection, models trained on different datasets exhibit incremental improvements in accuracy. Model_92_acc, trained on 60,000 audio samples, achieves the highest accuracy of 92.34%, showcasing the correlation between dataset size and detection accuracy. Similarly, for image deepfake detection, the accuracy of models increases with the size of the training dataset. The proposed model, leveraging ViT architecture on web-scrapped data, attains an accuracy of 89.35%, demonstrating its effectiveness in identifying image deepfakes. These findings highlight the importance of dataset size and architecture selection in enhancing the accuracy of deepfake detection systems. These key points highlight the effectiveness of SecureVision in providing accurate deepfake detection, thereby contributing to the mitigation of misinformation and fraudulent activities in various domains.

Figure 3 displays two line graphs, with the left graph showing training and validation accuracy and the right graph depicting training and validation loss over 50 epochs. In the accuracy graph, both training and validation accuracy increase over time, indicating learning progression, with training accuracy consistently higher. The loss graph shows an initial sharp decrease in training loss, which levels off, while validation loss experiences fluctuations, suggesting potential overfitting or instability in the validation phase. These graphs are instrumental in evaluating the model’s performance, balancing, and generalization capabilities over multiple training iterations. If machine learning [5] models are applied properly and iteratively, their accuracy can surpass 95% [4,7,53]. This can enhance the quality of outcomes [6] across several domains such as image processing [54], technology [5,55], and medicine [56].

Figure 4 The image shows a confusion matrix for a binary classification task, distinguishing between ‘Fake’ and ‘Real’ predictions. The matrix reveals the number of true positives (TP), true negatives (TN), false positives (FP), and false negatives (FN) generated by the classification model. The model accurately identified 2500 (2.5 × 10^2^) fake items (TP) and 3200 (3.2 × 10^2^) real items (TN), while misclassifying 42 real items as fake (FP) and 42 fake items as real (FN). The color gradient represents the frequency of predictions, with darker shades indicating higher numbers. This matrix is critical for assessing the model’s classification accuracy and its ability to generalize across different classes.

Figure 5 shows a confusion matrix for a binary classification task, distinguishing between ‘Fake’ and ‘Real’ predictions. The matrix reveals the number of true positives (TP), true negatives (TN), false positives (FP), and false negatives (FN) generated by the classification model. The model accurately identified 4731 fake items (TP) and 4728 real items (TN), while misclassifying 30 real items as fake (FP) and 32 fake items as real (FN). The color gradient represents the frequency of predictions, with darker shades indicating higher numbers. This matrix is critical for assessing the model’s classification accuracy and its ability to generalize across different classes.

#### 6.3.3. Comprehensive Analysis

(i)Efficacy: SecureVision uses cutting-edge deep learning methods, such as self-supervised learning and Vision Transformer models, to exhibit excellent efficacy in deepfake detection. Due to its multi-modal analysis technique, which enables it to process both audio and image data efficiently, it excels in properly identifying modified audio and visual information. SecureVision is able to handle large datasets and identify even minute changes in media content because of the inclusion of big data analytics. This strong strategy plays a major role in its ability to successfully defend digital media’s authenticity and integrity against emerging deepfake threats.(ii)Scalability: SecureVision can handle the increasing amount and complexity of digital content found in real-world situations. In order to handle enormous datasets without sacrificing efficiency, it does this by combining big data principles and sophisticated data processing skills. SecureVision can consistently adjust to the most recent methods of deepfake production because of the system’s architecture, which facilitates the integration of new data sources and developing technologies. SecureVision may be deployed in a variety of settings, from small-scale personal use cases to extensive organizational applications, owing to its flexibility and scalability, which is essential for real-time detection and reaction.(iii)Ethical Considerations: Significant ethical questions are brought up by the creation and application of SecureVision, mainly in relation to data security and privacy. Strong cyber security features like multi-factor authentication and encrypted data processing are included in SecureVision to reduce these worries, protect user data, and guarantee the appropriate application of detecting technology. Furthermore, the ethical ramifications of identifying and disclosing deepfakes are thoroughly examined, stressing the significance of accountability and openness. By balancing the protection of individual rights with effective deepfake detection, SecureVision seeks to uphold ethical norms in digital media integrity and cybersecurity.

#### 6.3.4. System Configuration

System Type: 64-bit operating system, x64-based processor

Processor: The processor of the setup is of Intel core 11th generation with Core™ i5- 1135G7 @2.40–2.42 GHz

RAM: The available RAM is 8 GB, which is also a minimum requirement

GPU: The GPU used is the Intel HD Graphics 620, which is an integrated GPU

Storage: The storage of the system is 512 GB SSD

Operating Systems: OS used is Windows 11 Home Single Language

## 7. Conclusions

The culmination of this research is a deepfake detection system employing a sophisticated neural network architecture tailored to discern the authenticity of audio and image data. For audio deepfake detection, the system integrates a proposed model achieving an outstanding accuracy of 92.34%, harnessing ASVspoof and multilingual datasets with the SpecRNet architecture. Similarly, for image deepfake detection, the system incorporates a proposed model trained on web-scrapped data using the ViT architecture, achieving an accuracy of 89.35%. This high accuracy underscores the system’s efficacy in identifying fabricated content and enhancing cybersecurity measures. Moreover, the system’s operational efficiency in moderate GPU resource environments ensures its suitability for real-time applications, marking a significant advancement in AI-driven cybersecurity.

## 8. Future Work

The ‘SecureVision’ initiative aims to broaden its algorithmic capabilities beyond facial and audio recognition, targeting full body deepfake detection. Emphasizing collaboration, ensemble methods, and algorithm transferability, it seeks to enhance detection accuracy and adaptability. Future efforts will focus on real-time detection and ethical implementation, aiming to establish a robust cybersecurity framework against deepfakes.

## Figures and Tables

**Figure 1 sensors-24-06300-f001:**
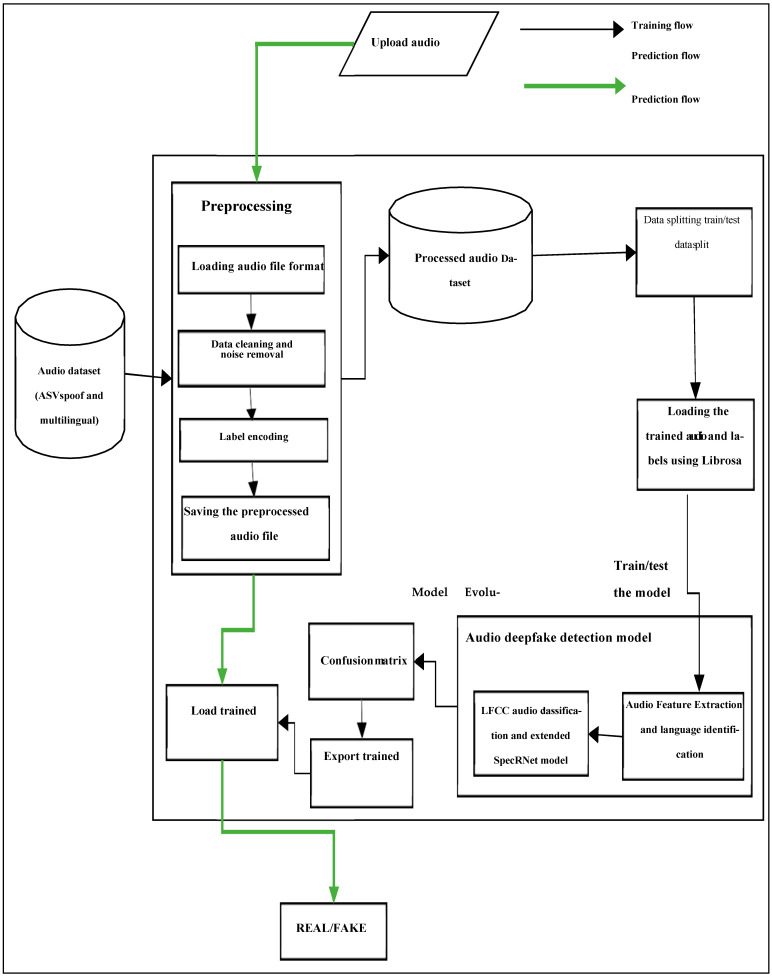
System architecture for audio deepfake detection.

**Figure 2 sensors-24-06300-f002:**
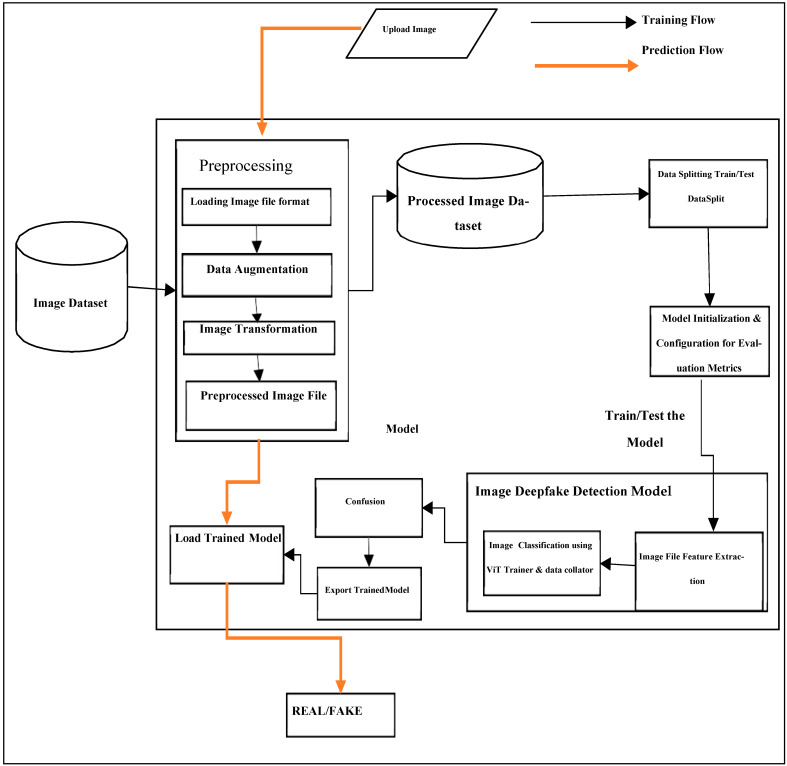
System architecture for image deepfake detection.

**Figure 3 sensors-24-06300-f003:**
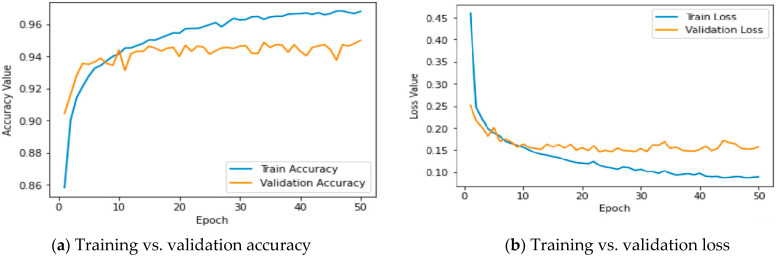
Graphs of training vs. validation based on accuracy and loss.

**Figure 4 sensors-24-06300-f004:**
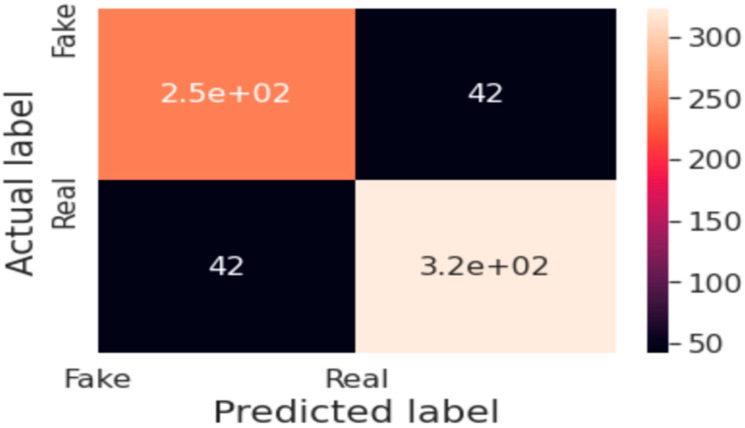
Confusion matrix of audio detection.

**Figure 5 sensors-24-06300-f005:**
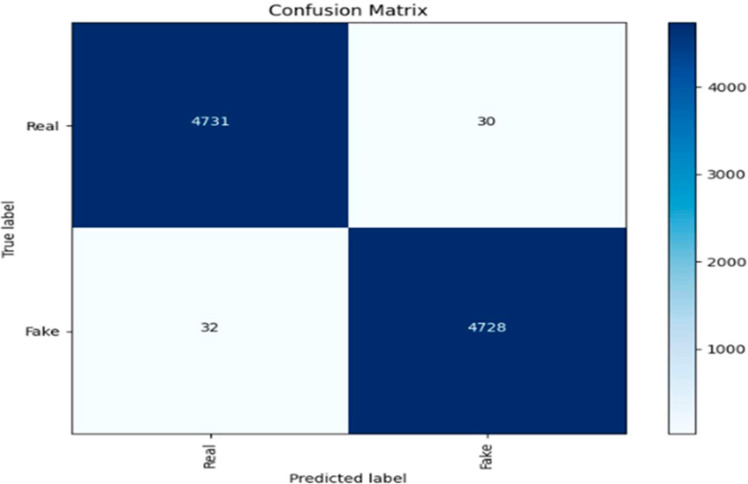
Confusion matrix for image detection.

**Table 1 sensors-24-06300-t001:** Audio-trained model accuracy.

S No.	Model Name	No. of Audios	Accuracy
1.	model_69_acc	30,000	69.00%
2.	model_85_acc	45,000	85.13%
3.	model_92_acc	60,000	92.34%

**Table 2 sensors-24-06300-t002:** Image-trained model accuracy.

S No.	Model Name	No. of Images	Accuracy
1.	model_73_acc	25,754	73.26%
2.	model_81_acc	38,956	81.44%
3.	model_89_acc	50,653	89.35%

**Table 3 sensors-24-06300-t003:** Comparison of various audio deepfake detection models.

S No.	Method	Dataset	Model	Claimed Performance	GPU Usage
1.	COVAREP + LSTM [34]	EMODB + SAVEE + CaFE	LSTM	Accuracy: 89.00%	High
2.	CAPTCHA [37]	Real-time 144p YT data	RNN + CNN	Accuracy: 71.00%	Moderate
3.	Proposed_Model	ASVspoof [21] + multilingual dataset	SpecRNet	Accuracy: 92.34%	Moderate

**Table 4 sensors-24-06300-t004:** Comparison of various image deepfake detection models.

S No.	Method	Dataset	Model	Claimed Performance	GPU Usage
1.	XceptionNet [43]	FaceForensics++ +Private web data	Traditional operator CNN	Accuracy: 96.00%	High
2.	FakeCatcher [39]	FaceForensics++	XceptionNet	Accuracy: 81.00%	Moderate
3.	Proposed_Model	Web Scrapped	ViT	Accuracy: 89.35%	Moderate

## Data Availability

Data are contained within the article.

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
