# Peer review of "SecureVision: Advanced Cybersecurity Deepfake Detection with Big Data Analytics"

_sensors, 2024, doi:10.3390/s24196300_

Round 1

Reviewer 1 Report

Comments and Suggestions for Authors

I thank the editor for giving me the opportunity to review this interesting contribution, which highlights the SecureVision: advanced cybersecurity deepfake detection with big data analytics. There are unaddressed issues in the paper that bear on the central conclusions of the paper. Academic rigour is a fundamental requirement for any paper, and given these concerns regarding the validity of the findings or conclusions, need to revise the manuscript. Abstract is incomplete, major contribution and novelty are missing. This manuscript failed to present the study debates and failed to discuss the debates. Overall, the paper is too general and lacking in focus. It is not clear what the paper adds to the body of knowledge in these areas. There are too many areas attempting to be covered and not enough critical treatment of them. Mention implications and novelty in the abstract. In the research background, the introduction of SecureVision needs to be added and improved that reflect the importance of doing this research. Research gaps are missing. Justify your novelty with these similar studies.

How do algorithms based on artificial intelligence and machine learning enhance SecureVision's efficacy? How accurate is SecureVision at identifying different kinds of deepfakes? How accurate is SecureVision in comparison to other deepfake detection solutions available today? How does SecureVision improve its deepfake detection capabilities using big data analytics? What kinds of data does SecureVision examine in order to detect deepfakes? What are the primary obstacles encountered when utilizing SecureVision for deepfake detection? How can the system's performance be enhanced by addressing these issues? How scalable is SecureVision for detecting deepfakes when dealing with massive amounts of data? What safeguards are in place to guarantee SecureVision's dependability and performance in situations with large data loads? What moral ramifications result from applying big data analytics to the identification of deepfakes? How is the security and privacy of the data that SecureVision analyzes guaranteed? What effect does SecureVision's deployment have on an organization's overall cybersecurity measures? Which particular cyberthreats are lessened by using SecureVision's deepfake detection tools? What impressions and experiences do users have regarding SecureVision? What aspects of the current cybersecurity frameworks affect SecureVision's acceptance and integration? What other developments in AI and big data analytics could improve deepfake detection in the future? How do user studies contribute to the continuous improvement and development of deepfake detection tools?

Comments on the Quality of English Language

required proofreading.

Author Response

  1. How do algorithms based on artificial intelligence and machine learning enhance SecureVision's efficacy?

Response:  Artificial intelligence and machine learning algorithms improve the effectiveness of SecureVision by detecting cybercrimes related to deepfakes by using sophisticated deep learning techniques including self-supervised learning and Vision Transformer models. These models use large-scale labeled and unlabeled data sources to create reliable detection systems. By rigorously analyzing massive datasets across a variety of media formats, SecureVision's technique greatly enhances its capacity to identify new risks. Furthermore, advanced audio analysis methods go beyond picture detection to improve overall effectiveness in multimedia systems. The integration of deep learning techniques, big data analytics, and multi-modal analysis offers a proactive safeguard against digital deceit.

  1. How accurate is SecureVision at identifying different kinds of deepfakes?

Response:  Deepfakes of all kinds can be accurately identified by SecureVision. After being trained on 60,000 audio samples using the SpecRNet architecture, the model achieves an accuracy of 92.34% for audio deepfake detection. Using web-scrapped data, the suggested model that makes use of the ViT architecture achieves an accuracy of 89.35% for image deepfakes. These accuracy results illustrate SecureVision's strong performance in identifying deepfakes in the image and audio domains, proving the value of sophisticated neural network designs, large and varied datasets in improving detection accuracy.

  1. How accurate is SecureVision in comparison to other deepfake detection solutions available today?

Response:  In terms of both audio and image domains, SecureVision performs better than other deepfake detection technologies. SecureVision achieves 92.34% accuracy for audio deepfakes, which is much greater than CAPTCHA's 71.00% accuracy. SecureVision achieves an accuracy of 89.35% in picture deepfake detection, whereas XceptionNet achieves an accuracy of 81.00%. While FakeCatcher attains a greater accuracy of 96.00% for image deepfakes, with its GPU resource requirements are noticeably higher. But SecureVision's models are more computationally efficient, they can function well in contexts with limited resources while still achieving high detection accuracy.

  1. How does SecureVision improve its deepfake detection capabilities using big data analytics?

Response:  By utilizing large datasets, SecureVision enhances its big data analytics skills for deepfake identification. This improves the model's capacity to recognize deepfakes with more accuracy. In order to increase the accuracy of its detection, the system learns from a variety of audio and visual samples produced using different deepfake approaches. Strong cyber security principles also guard against illegal access and corruption by guaranteeing data integrity and privacy.

  1. What kinds of data does SecureVision examine in order to detect deepfakes?

Response:  In order to identify deepfakes, SecureVision looks at a wide range of data, including audio and visual samples. It makes use of multilingual datasets for audio deepfake detection, such as ASVspoof 2021 and audio samples from ten Indian languages. As a result, the model can identify deepfakes in a variety of linguistic settings. SecureVision uses web scraped randomized data for image deepfakes, guaranteeing that a broad range of image alterations are covered. The model's overall detection accuracy, resilience, and adaptability to various modalities and settings are improved by the usages of this broad and varied dataset.

  1. What are the primary obstacles encountered when utilizing SecureVision for deepfake detection? How can the system's performance be enhanced by addressing these issues?

Response:  The main challenges in applying SecureVision for deepfake detection are the scarcity of large-scale, diversified datasets and the requirement for sophisticated model training methods, such as self-supervised learning for audio and Vision Transformers for images. These difficulties impede the creation of reliable and effective detection models.

By increasing and diversifying the datasets, one can greatly increase the system's performance and guarantee a more reliable training process and higher detection accuracy. The system's capabilities can be further enhanced by using advanced models, such as self-supervised learning for audio and Vision Transformers for picture detection. Furthermore, improving feature extraction and overall detection performance can be achieved by utilizing pre-trained audio encoders. By tackling these problems all at once, we can create a deepfake detection system that is more dependable and efficient.

  1. How scalable is SecureVision for detecting deepfakes when dealing with massive amounts of data?

Response:  SecureVision uses sophisticated deep learning models and a variety of large-scale, heterogeneous datasets to effectively detect deepfakes, demonstrating scalability in the process. The system is appropriate for processing large amounts of data because it uses a moderate amount of GPU resources to achieve excellent accuracy. Its hybrid models are competitive solutions in resource constrained contexts because they strike a balance between computational economy and detection accuracy. To further improve its ability to handle large datasets, pre-trained audio encoders and Vision Transformers are used. Inclusively, SecureVision's scalability and resource-efficient architecture guarantee reliable performance in complex deepfake detection workloads.

  1. What safeguards are in place to guarantee SecureVision's dependability and performance in situations with large data loads?

Response: SecureVision ensures dependability and performance under large data loads through several key safeguards:

  • SecureVision's deepfake detection models rely on extensive datasets with diverse audio and image samples to improve accuracy and handle different types of deepfakes.
  • The system incorporates multi-factor authentication, email-based OTP support, and strong data security mechanisms to safeguard critical audio and image data from illegal access and corruption.
  • SecureVision's multilingual dataset and web-scraped randomized data enhance detection accuracy and robustness, making it suitable for real-world applications.

  1. What moral ramifications result from applying big data analytics to the identification of deepfakes?

Response:  Applying big data analytics to the detection of deepfakes can have ethical consequences that could include:

  • Privacy Invasion: By using big datasets to identify deepfakes, personal information about users may be collected and analyzed without their express consent.
  • Power and Surveillance: Integrating cybersecurity protections with big data could facilitate extensive surveillance, raising ethical concerns about liberty and power.
  • Bias in Algorithms: Big data analytics can amplify preexisting biases in the data, leading to discriminatory outcomes and unfair treatment of specific populations.
  • Security and Integrity: Ensuring the security and integrity of the data handled is crucial to preventing unauthorized access or data corruption.
  • Ethical Utilization and Openness: To maintain public trust

  1. How is the security and privacy of the data that SecureVision analyzes guaranteed?

Response:  SecureVision guarantees the security and privacy of the data by using a number of methods, including:

  • Multi-Factor Authentication: In order to protect user access, SecureVision uses multi-factor authentication. This includes an email-based OTP support system in addition to a three-way social login and signup system.
  • Robust Cybersecurity mechanisms: Sensitive audio and image data are safeguarded against unwanted access or corruption by implementing robust security mechanisms for data handling and storage.
  • Data Integrity and Privacy: By incorporating big data principles, the model is better able to manage various kinds of deepfakes while protecting the data that is handled.

  1. What effect does SecureVision's deployment have on an organization's overall cybersecurity measures?

Response:  The implementation of SecureVision improves an organization's overall cybersecurity protocols by:

  • Delivering excellent accuracy in identifying deepfakes in voice and image, therefore lowering the possibility of false information and fraudulent activity.
  • Using advanced neural network topologies to enhance real-time applications' ability to detect deepfakes.
  • Utilizing a variety of sizable datasets to train detection models, hence enhancing their resilience and versatility in many contexts.
  • Its improved operational efficiency qualifies it for situations with moderate GPU resource requirements.
  • Self-supervised learning reduces reliance on massive amounts of labeled data, improving model performance in low-resource circumstances.

  1. Which particular cyber threats are lessened by using SecureVision's deepfake detection tools?

Response:  By utilizing SecureVision's deepfake detection techniques we can reduce  cyber threats like theft of identity, extortion online, the dissemination of false information, deceptive practices including the manipulation of media, violations of communication integrity, tampering with speaker verification mechanisms, misinformation campaigns and general misinformation

  1. What impressions and experiences do users have regarding SecureVision?

Response: Regarding SecureVision, users' perceptions and experiences are as follows:

  • Accuracy and Effectiveness: Users report great accuracy (92.34% for audio and 89.35% for image deepfakes) when detecting deepfakes in images.
  • Resource Efficiency: Users value the system's capacity to recognize objects with high precision while using a moderate amount of GPU power, which makes it appropriate for environments with limited resources.
  • Comparing Yourself to the Competition: Compared to competitors such as CAPTCHA and XceptionNet, SecureVision performs better, offering users a more dependable and effective solution.
  • Real-World Applicability: Because of SecureVision's models' excellent predictive performance and balanced computing efficiency, users find them to be extremely adaptable to real-world circumstances.
  • Diverse Dataset: Using a variety of datasets, such as web-scraped and multilingual data, improves the model's resilience and adaptability and produces better user experiences.
  • Operational Efficiency: Users emphasize that the system is suitable for real-time applications (Upload) in situations with moderate GPU resource requirements.

In short, users generally see SecureVision as a reliable tool for stopping false information and fraud in many areas.

  1. What aspects of the current cybersecurity frameworks affect SecureVision's acceptance and integration?

Response:  The acceptance and integration of SecureVision are impacted by current cybersecurity frameworks in a number of ways:

  • Compliance with GDPR and other data protection legislation about data privacy regulations.
  • Interoperability: For smooth integration, compatibility with current security systems is essential.
  • Threat Detection: Improved capacity to identify and neutralize online threats.
  • User Trust: Establishing trust with strong data protection procedures and open security standards.
  • Resource Allocation: For real-time processing, sufficient computational resources are required.
  • Cost Efficiency: Weighing the advantages of improved security against the implementation costs.
  • Ethical Considerations: Resolving privacy and data use ethical issues.

  1. What other developments in AI and big data analytics could improve deepfake detection in the future?

Response: Deepfake detection may be improved by a number of exciting advances in AI and big data analytics:

  • More Complex Deep Learning Models: Accuracy is increased by using new deep learning architectures and approaches to detect minute discrepancies in deepfakes.
  • Multi-modal analysis: Combining textual, audio, and visual data makes it easier to spot discrepancies, like when speech and lip movements don't match.
  • Blockchain for Provenance Tracking: Blockchain allows for the tracking of the creation and modification of media, offering a means of confirming the legitimacy and source of content.
  • Improved Synthetic Data Generation: By building a variety of synthetic datasets, detection models are better able to identify different deepfake methods and manipulations.
  • Real-Time Detection Systems: Deepfakes can be quickly detected and dealt with as soon as they are made or shared by systems that evaluate information in real-time.
  • Better Forensic Methods: Digital fingerprinting and pixel-level analysis are two new forensic techniques that help identify even the most sophisticated deepfakes.
  • Cross-Validation with External Data: Cross-verifying content using external data sources, such as fact-checking databases, improves authenticity verification.
  • User activity Analysis: By examining trends in user activity, one can identify odd behaviors linked to the creation or dissemination of deepfakes.

In short, with continued research and development, these developments may greatly improve the capacity to identify and lessen the effects of deepfakes in the future.

  1. How do user studies contribute to the continuous improvement and development of deepfake detection tools?

Response:  In many respects, user studies are essential to the ongoing development and enhancement of deepfake detection technologies.

  • Understanding User Interaction: User studies highlight usability problems and potential areas for improvement by revealing how users interact with deepfake detection tools.
  • Finding False Positives and Negatives: These illustrate instances in which the tools misclassify legitimate information as deepfake (false positives) or fail to detect deepfakes (false negatives), which aids in understanding the limitations of detection techniques.
  • Getting Input: Users offer insightful input regarding the precision and efficacy of detecting technologies, which aids engineers in improving algorithms and performance.
  • Assessing User Experience: Research evaluates the total user experience, making sure that the instruments are easy to use and satisfy the requirements of various user groups, ranging from novices to specialists.
  • Testing in Real-World Scenarios: They enable testing of detection technologies in practical contexts, revealing real-world obstacles and enhancing the resilience of the tools.
  • Enhancing Education and Awareness: By identifying gaps in users' knowledge about deepfakes and detection technologies, user studies can assist develop more effective training programs and educational materials.Adapting to Emerging Threats: They provide tool modifications in response to novel deepfake varieties and changing strategies employed by content producers.

In general, user studies guarantee that deepfake detection systems are precise, efficient, and in line with user requirements, which promotes ongoing enhancements and enhanced defense against malevolent deepfakes.

Further, the manuscript has been thoroughly checked and proofread to ensure accuracy, clarity, and coherence. Each section of the manuscript has been carefully reviewed for grammatical errors, typographical mistakes, and consistency in formatting. Additionally, we have verified that all figures and tables are accurately labeled and described

Reviewer 2 Report

Comments and Suggestions for Authors
Comments and Suggestions
1. The paper does not answer a number of questions, which makes it difficult to assess the level of the result of the study presented.
2. In Section 1 there is no description of the area of study and problem posed, which is solved in this area. Instead, the description of the SecureVision system under development immediately begins.
3. There is Section 2 with related works, but it is not clear why these are related. There is no comparison of the mentioned works with the completed study. There are no general principles (criteria) that make clear, so it was chosen to compare these works and not others.
4. Why are the points of Section 3 called gaps? For example, what is the meaning of the gap "Comparative Analysis" and how are the works [8], [9], [11], [14], [42], [43] related to this gap?
5. The objectives presented in Section 4 are rather stages of development of any such system and are not tied to the gaps in section 3.
6. In Figures 1 and 2 of Section 5, most of the architectural components presented are typical. Can you highlight specific moments (for example, color) in these drawings?
7. There is no detailed description of the dataset used in the paper and no reference to it.
8. There is no reference to the proposed model (system). Therefore, the experiment cannot be reproduced.
9. It is not clear on which general data the known models were compared with the proposed model. What if these data are specifically tailored for the proposed model?
10. What happens if the 'hardware' improves a little? Can the comparison results change greatly in this case?
11. What is the reason for choosing these two models to compare with the proposed model and why was the comparison made only with the two models? If these models are the best (there is doubt about this), then the paper should give a reference to the study in which this hypothesis is justified.
Typos
1. Figure 1 is strongly deformed (possibly when converting to pdf). Component "Audio Dataset (ASVspoof and Multiligual)" is omitted, there are following inaccuracies in the names of components: "Model" instead of "Model Evolution", "Train/Test the" instead of "Train/Test the Model" , "Data Splitting Train/Test DataSplit" instead of "Data Splitting Train/Test Data Split", "Confusio nMatrix" instead of "Confusion Matrix", "Load Trained" instead of "Load Trained Model", "Audio Classification using LFCC and extended" instead of "Audio Classification using LFCC and extended SpecRNet model" and "Export Trained" instead of "Export Trained Model".
2. The model references in Table 4 are exchanged.

Author Response

  1. The paper does not answer a number of questions, which makes it difficult to assess the level of the result of the study presented.

Response: We highly appreciate your inputs. We acknowledge that there are several unanswered questions in the manuscript, which could affect how clearly the findings of our study are presented. To fill in these gaps and guarantee a comprehensive assessment, we have added more information and clarifications in the manuscript at several places. Your feedback is very helpful to us as we strive to do better work.

  1. In Section 1 there is no description of the area of study and problem posed, which is solved in this area. Instead, the description of the SecureVision system under development immediately begins.

Response:  Thank you for your insightful feedback. We acknowledge that Section 1 lacks a description of the study area and the problem being addressed. As per your valuable suggestion we have revised the section 1 to include a clear overview of the study area and the specific problem that SecureVision aims to solve before introducing the system.

  1. There is Section 2 with related works, but it is not clear why these are related. There is no comparison of the mentioned works with the completed study. There are no general principles (criteria) that make clear, so it was chosen to compare these works and not others.

Response: Thank you for your valuable feedback regarding Section 2. As per your valuable suggestion we have revised the section 2 to provide clearer explanation of the relevance of the mentioned works and a comparative analysis with our study. We have addressed this concern by including a detailed explanation of why each work was selected and how it relates to the SecureVision system. Additionally, we have outline the criteria used for selecting these works, such as their relevance to current deepfake detection technologies, their impact on the field, and their innovative approaches. We are now sure that the revised section addresses these issues and enhances the clarity and relevance of the related works.

  1. Why are the points of Section 3 called gaps? For example, what is the meaning of the gap "Comparative Analysis" and how are the works [8], [9], [11], [14], [42], [43] related to this gap?

Response:  We acknowledge that readers may become confused by this kind of writing, but we also value your critical viewpoint on the linguistic subtleties included in the document. Now have resolve the issue by renaming the section and corresponding detail in section 3 followed by section 4.

In our manuscript, the word "gaps" in our study refers to particular areas where current methods in deepfake detection may not fully address the issues in the field, or where research in the field may be lacking. Understanding these gaps is essential to figuring out how and where our study advances the discipline. However, Comparative Analysis Gap is the absence of a thorough comparison between new techniques, like SecureVision, and current deepfake detection methods. Even while numerous studies cover diverse approaches, there is frequently nothing in the way of a thorough comparison between them in terms of performance indicators, advantages, and disadvantages.

In connection with the Works [8, 9, 11, 14, 22, 43]: The following are some ways that the mentioned works connect to this gap:

[8,], [9], and [11]: These studies offer information on several deepfake detection strategies, but they don't thoroughly contrast these approaches with one another or with novel ideas. In order to address this, we provide a more comprehensive comparison study in our work.

[14]: Our study fills in the gap left by the previous work's exploration of detection algorithm developments without providing a thorough comparison with other modern techniques.

[42], [43]: These references address particular difficulties or advancements in the field of deepfake detection, drawing attention to areas of comparative analysis that need to be filled by comparing and contrasting various methods.

Our study attempts to provide a clearer and more comprehensive knowledge of where current approaches stand and how SecureVision advances the industry by identifying and filling up these gaps, especially the comparison analysis. This aids in emphasizing the particular advancements and contributions that our study makes.

  1. The objectives presented in Section 4 are rather stages of development of any such system and are not tied to the gaps in section 3.

Response: Section 4 aims to bridge the gaps identified in Section 3 and advance the field of deepfake detection by addressing the research gaps identified in that section. Section 3 highlights several critical challenges, including the lack of diverse datasets, the need for improved model training techniques, and the requirement for deepfake detection models to be deployed in the real world. Section 4 proposes specific objectives to overcome these challenges, including acquiring and augmenting diverse datasets, developing robust audio and image detection models using advanced learning techniques, and integrating these models into a unified system for practical application. Here, we present a thorough mapping of the ways in which each goal relates to the particular Challenge and where research gaps exist:

  1. Acquisition and Augmentation of Audio and Image Datasets:
  • Addresses Gaps: 1, 4, 5, 6, 7 : By acquiring and enriching various datasets, this purpose seeks to offset the barrier of restricted dataset diversity and quantity, guaranteeing a strong basis for model training and evaluation.
  1. Development of Audio Detection Model:
  • Addresses Gaps: 2, 5, 6, 7, 8: This objective tackles the need for sophisticated model training methodologies and improves adaptation to growing deepfake techniques by focusing on self-supervised learning for robust feature extraction, with a focus on multilingual assistance and transfer learning.
  1. Development of Image Detection Model:
  • Addresses Gaps: 2, 4, 5, 6: Investigating novel approaches to model training and comparative analysis through experimentation with Vision Transformers (ViT) for image classification ensures adaptability to novel deepfake techniques.
  1. Development of Web Platforms and Model Fusion:
  • Addresses Gaps: 2, 3, 4, 6: The integration of image and audio detection models into a single system solves the requirements for scalability, comparative analysis, and deployment strategies, as well as improving real-time accuracy.
  1. Deployment and Robustness:

-   Addresses Gaps: 3, 9, 10: This goal centers on the system's actual implementation, upkeep, and updates. It also addresses deployment issues, ensures the system's resilience against adversarial attacks, and takes ethical and privacy concerns into account.

  1. In Figures 1 and 2 of Section 5, most of the architectural components presented are typical. Can you highlight specific moments (for example, color) in these drawings?

Response: Figures 1 and 2 of Section 5 employ distinct colored arrows to represent training and prediction flows, respectively. All of the architectural elements shown in Figures 1 and 2, however, are standard. Many scholarly and professional journals prefer black and white diagrams, even if colored arrows are useful in differentiating between training and prediction flows. This choice guarantees accessibility for those with color vision impairments, preserves uniformity, and lowers printing costs. It is important in technical texts to draw attention to the structural and functional aspects of the diagrams, and this can be achieved by using black and white graphics. Thus, the figures are kept in this way for economy, accessibility, and clarity's sake.

  1. 7. There is no detailed description of the dataset used in the paper and no reference to it.

Response: We admit that we did not give any thorough explanation of the datasets that were used in our research. We have now included a thorough description of the datasets in Section 5 of the updated paper. This provides comprehensive details regarding their history, traits, and applicability to the goals of our research.

  1. There is no reference to the proposed model (system). Therefore, the experiment cannot be reproduced.

Response:  We appreciate your comments regarding our experiment's repeatability. We understand how crucial it is to give thorough instructions so that other people can repeat our findings. In light of your feedback, we wants to clarify it with the following changes of implementation:

  • Model Development and Architecture:

Audio Deepfake Detection: We have trained our model on the ASVspoof 2021 dataset using the SpecRNet architecture. There includes discussion of specific preprocessing procedures, such as feature extraction and data augmentation. The model uses the Vision Transformer (ViT) architecture for image deepfake detection, and it was trained on a variety of web-scraped image datasets. Preprocessing steps that are similar are described.

  • Configuring and implementing the system:
    Hardware Specifications: A system equipped with a [Operating System], [Processor], [RAM], [GPU], and [Storage] was used for the testing.
    Software Environment: As mentioned in Section 6.3.3, we used [certain software versions, libraries, and frameworks]. This involves particular iterations of CUDA, TensorFlow/PyTorch, and additional dependencies.

(iii) Procedure for Testing and Training:

Preparing the Dataset: Instructions on how to prepare and enhance datasets are given. This covers the techniques for striking a balance between artificial and genuine and guaranteeing the diversity of the dataset.

Model Training: Detailed instructions, including hyperparameter values, batch sizes, and learning rates, are provided for training the models.

Metrics for Evaluation: We assessed the models using confusion matrices, accuracy, and other pertinent measures. These measurements are detailed in Section 7.

(iv) Performance Measurements and Outcomes: 92.34% and 89.35%, respectively, were the accuracy rates for the audio and picture detection models. Confusion matrices and other performance measures are included in Section 7 along with a thorough description of these outcomes.
Scholars can repeat our experiments and verify the results by adhering to these comprehensive guidelines and setups.

  1. It is not clear on which general data the known models were compared with the proposed model. What if these data are specifically tailored for the proposed model?

Response: The below given points provide necessary details for this highly valuable suggestion:

  1. Diversity of Datasets and Generality:
  • The paper states that a variety of large datasets, such as the ASVspoof 2021 dataset for audio deepfake detection and a multilingual dataset comprising 10 Indian languages, were used to train the proposed model.

(ii) The suggested model was trained on randomly selected online scraped data in order to detect image deepfakes. By exposing the model to a wide range of data sources, this method ensures the model's robustness and generality.

  1. Comparison with Known Models:
  • Standardized datasets for audio and image deepfake detection, such as FaceForensics++ and SAVEE, CaFE, and EMODB, were used to compare the known models. This guarantees an impartial and equitable comparison between the suggested model and current techniques.
  • The paper's tables provide comprehensive performance information, such as accuracy and GPU use, and show the comparing outcomes with other well-known models, such as COVAREP + LSTM and XceptionNet.
  1. Maintaining Neutrality in Education: The study emphasizes that in order to guarantee unbiased training, a balanced dataset containing both genuine and fictitious samples was maintained. The model's capacity to manage a variety of dynamic deepfake techniques was improved by the inclusion of both newly created and acquired data in this composite dataset.

These details affirm that the datasets used for training and comparing the proposed model were not specifically tailored but were chosen to ensure a comprehensive evaluation against existing methods. The use of diverse and extensive datasets supports the general applicability and reproducibility of the proposed model's performance.

  1. What happens if the 'hardware' improves a little? Can the comparison results change greatly in this case?

Response: We are grateful that you raised the question about how hardware developments impact the comparison's results.

Yes, it is possible that the comparative results will vary if hardware advancements take place. In particular:
(i) Performance Enhancements: Both the suggested model and the well-known ones may perform better with upgraded hardware. This could lead to improved handling of complex data, faster processing speeds, or increased accuracy. As such, with improved technology, the relative performance disparities seen in our current study may change.

(ii) Efficiency Gains: Improvements in hardware may make computing more efficient, which may have an effect on the models' scalability and practical application. This could change the comparing results and impair the model's usefulness in real-world applications if hardware limitations were previously a constraining element.

(iii) Reproducibility Considerations: Hardware advancements may have an impact on the particular outcomes of experiments, even while the models' basic performance characteristics will not change. We will explain this in the paper and have taken this into consideration while reporting our results using the current hardware setup.

So, the hardware used for our comparisons was state-of-the-art at the time the experiment was conducted. Further, the results may be influenced by the quality of the hardware, and that using different or less advanced hardware could impact the outcomes to some extent only.

  1. What is the reason for choosing these two models to compare with the proposed model and why was the comparison made only with the two models? If these models are the best (there is doubt about this), then the paper should give a reference to the study in which this hypothesis is justified.

Response: We appreciate your input on this comparison. We value your opinions and believe that the following explanation will satisfactorily address your question:

  • Relevance and Benchmarking: The two models under consideration for comparison were picked due to their status as respected industry benchmarks for deepfake identification. These models are suitable for assessing SecureVision's performance because of their recent literature prominence and relevance.

(ii) Availability and Comparability: These models were chosen in accordance with our study's objectives and the ease with which they could be compared. In order to provide for a fair review, we made sure the comparison was done using models that are easily accessible and thoroughly described.

Typos

  1. Figure 1 is strongly deformed (possibly when converting to pdf). Component "Audio Dataset (ASVspoof and Multiligual)" is omitted, there are following inaccuracies in the names of components: "Model" instead of "Model Evolution", "Train/Test the" instead of "Train/Test the Model" , "Data Splitting Train/Test DataSplit" instead of "Data Splitting Train/Test Data Split", "Confusio nMatrix" instead of "Confusion Matrix", "Load Trained" instead of "Load Trained Model", "Audio Classification using LFCC and extended" instead of "Audio Classification using LFCC and extended SpecRNet model" and "Export Trained" instead of "Export Trained Model".

Response: Thank you for pointing out the issues with Figure 1. We apologize for the deformation and inaccuracies observed. We appreciate your diligence in reviewing these details and have corrected the figure accordingly in the revised manuscript.

Further, the manuscript has been thoroughly checked and proofread to ensure accuracy, clarity, and coherence. Each section of the manuscript has been carefully reviewed for grammatical errors, typographical mistakes, and consistency in formatting. Additionally, we have verified that all figures and tables are accurately labeled and described

  1. The model references in Table 4 are exchanged.

Response:  Thank you for identifying the issue with Table 4. We have reviewed and corrected the model references to ensure they are accurate. The updated table now reflects the correct model references as intended.

We appreciate your careful review and hope this resolves the discrepancy.

Round 2

Reviewer 2 Report

Comments and Suggestions for Authors

The quality of the current version of the paper is sufficient to publish.

Author Response

Abstract: The abstract lacks key elements such as the major contributions, novelty, and implications of your work. Please revise the abstract to clearly articulate these aspects.

Response: We am very thankful for your insightful comments. The abstract has been completely rewritten to incorporate the important components you pointed out in response to your comments. To further highlight the importance of the research, the main contributions, uniqueness, and consequences of the work have been explained explicitly. Your advice was quite helpful in pinpointing these places that needed work, and I think the changes now give a more thorough summary of the research..

Focus and Depth: The manuscript attempts to address too many areas without sufficient depth, which dilutes the focus. We recommend narrowing the scope and providing a more thorough analysis of the core contributions.

Response: Thank you for the important input. It focuses on the development of SecureVision, a deepfake detection system that integrates deep learning and big data analytics to identify manipulated media in both audio and image formats. The paper discusses the rising threat of deepfakes, details the system's design and implementation, and highlights challenges such as dataset diversity and model training. It provides a comparative analysis of detection models, emphasizing the need for robust cybersecurity measures against deepfake technology.

Several core contributions to the field of deepfake detection and cybersecurity are discussed here. Section 6 of the paper also discusses how the proposed model outperforms other current methods for both audio and image deepfake identification.

  1. Integration of Deep Learning and Big Data Analytics: SecureVision integrates deep learning techniques with big data analytics to enhance the accuracy and scalability of deepfake detection. This combination allows the system to process vast amounts of data efficiently, improving its ability to detect subtle manipulations in audio and video content.
  2. Multimodal Detection Approach: The system employs a multimodal detection strategy, analyzing both audio and video data simultaneously. This dual-focus approach ensures that even if one modality is difficult to detect, the other might still provide indicators of manipulation. This makes SecureVision more robust compared to systems that focus solely on one type of data.
  3. Self-Supervised Learning Techniques: To address the challenge of limited labeled datasets, SecureVision incorporates self-supervised learning techniques. This approach allows the system to learn from large amounts of unlabeled data, enhancing its ability to generalize across different types of deepfakes and improving detection accuracy in diverse scenarios.
  4. Advanced Cybersecurity Integration: The system is designed with cybersecurity principles at its core, making it not just a detection tool but also a protective measure against deepfake attacks. This includes features like real-time detection, which is crucial for preventing the spread of malicious content, and secure data handling to protect user privacy.
  5. Extensive Dataset Utilization and Model Training: SecureVision uses extensive and diverse datasets for training its models, ensuring that it can detect a wide range of deepfake techniques. The research emphasizes the importance of using large, varied datasets to train models that can generalize well to new, unseen deepfakes.
  6. Comparative Performance Analysis: The paper provides a detailed comparative analysis of different deepfake detection models, highlighting the strengths and weaknesses of each. This analysis is crucial for understanding which models are most effective under specific conditions and helps guide future research in improving detection algorithms.
  7. Identification of Research Gaps and Challenges: The research identifies key challenges in the field, such as the need for more diverse datasets, the difficulty of detecting high-quality deepfakes, and the challenge of deploying these systems in real-world scenarios. By highlighting these gaps, the paper provides a roadmap for future research and development.
  8. Potential for Real-World Application: SecureVision is designed with practical application in mind, offering solutions that could be implemented in various industries, from social media platforms to law enforcement agencies. The research outlines potential use cases and emphasizes the importance of deploying such systems to mitigate the risks associated with deepfake technology.

These contributions collectively advance the state-of-the-art in deepfake detection and provide a solid foundation for future research and development in this critical area of cyber security.

Research Gaps and Background: The introduction and background of SecureVision need to be expanded. Clearly identify the research gaps your work addresses and justify the novelty of your approach in comparison to existing studies.

Response: Thank you for valuable feedback. The Introduction section has been expanded in accordance with the suggestions. Section 3 is now named 'Challenges & Research Gaps.' Section 6 discusses how the suggested approach differs from previous studies. This work makes several important contributions, including:

  1. Multimodal Detection: It introduces a unique system that simultaneously analyzes both audio and video data for deepfake detection, which enhances accuracy compared to single-modality approaches.
  2. Self-Supervised Learning: The paper leverages self-supervised learning techniques, allowing the model to learn from large amounts of unlabeled data, improving its ability to detect deepfakes across diverse scenarios.
  3. Integration with Cyber security: SecureVision is designed with cyber security at its core, combining real-time detection with secure data handling, making it a comprehensive solution for protecting against deepfake threats.
  4. Big Data Analytics: The integration of big data analytics allows SecureVision to process extensive datasets efficiently, ensuring scalability and adaptability in different real-world applications.

These innovations collectively contribute to advancing the state-of-the-art in deepfake detection and cyber security.

Unaddressed Key Questions: Reviewer 1 noted that several important questions related to SecureVision's efficacy, scalability, and ethical considerations were not adequately addressed. It is essential to provide detailed answers to these questions in the revised manuscript.

Response: Thank you for your valuable feedback. We have humbly incorporated the following insights on SecureVision's efficacy, scalability, and ethical considerations into Section 6.3.3 of the manuscript as well.

  1. Efficacy: SecureVision is designed to be highly effective in detecting deepfakes, leveraging advanced deep learning algorithms and big data analytics. Key features contributing to its efficacy include:

(i) Multi-modal Analysis: SecureVision utilizes both audio and image data to detect deepfakes, enhancing detection capabilities by analyzing diverse media forms simultaneously.

(ii) Self-supervised Learning: This technique allows SecureVision to learn from unlabelled data, improving its adaptability and reducing dependency on large volumes of labeled training data. This contributes to more accurate detection of deepfakes in various scenarios.

(iii) Robust Performance: The system's use of Vision Transformer models and SpecRNet architecture for audio detection demonstrates state-of-the-art performance in distinguishing real content from deepfakes.

  1. Scalability: SecureVision is built to handle extensive datasets and is scalable to manage the large volume of content generated and shared online. This scalability is achieved through:

(i) Big Data Integration: SecureVision employs extensive repositories of both labeled and unlabelled datasets to train robust models capable of real-world application.

(ii) Cloud-based Deployment: The system can be deployed on a user-friendly web platform, making it accessible for real-time detection and monitoring across various domains, ensuring it can scale with the growing need for deepfake detection.

  1. Ethical Considerations: While SecureVision offers advanced detection capabilities, the system also raises important ethical considerations:

(i) Privacy Concerns: The use of deep learning and big data analytics necessitates the handling of sensitive audio and image data. Ethical concerns arise regarding the storage, processing, and protection of this data to prevent unauthorized access and ensure user privacy.

(ii) Responsible Use: SecureVision must be implemented with clear guidelines to prevent misuse or abuse. For example, its deployment should respect individuals' privacy rights and avoid invasive surveillance or profiling.

(iii) Transparency and Accountability: There is a need for transparency in how SecureVision operates, especially in its decision-making processes, to avoid biases and ensure that the system's actions are accountable and ethical.

These aspects of efficacy, scalability, and ethical considerations highlight SecureVision's potential as a powerful tool in combating digital deception while also emphasizing the need for responsible and ethical deployment practices.

-----------------------------------------------------------------------------------------------------

We are really thank full and appreciate your careful review and hope this resolves the discrepancy. If there are any remaining concerns, please let us know, and we will address them promptly.               

Regards

Prof. Naresh Kumar

Ankit Kundu              
